# Tumorspheres as In Vitro Model for Identifying Predictive Chemoresistance and Tumor Aggressiveness Biomarkers in Breast and Colorectal Cancer

**DOI:** 10.3390/biology13090724

**Published:** 2024-09-15

**Authors:** Toni Martinez-Bernabe, Pere Miquel Morla-Barcelo, Lucas Melguizo-Salom, Margalida Munar-Gelabert, Alba Maroto-Blasco, Margalida Torrens-Mas, Jordi Oliver, Pilar Roca, Mercedes Nadal-Serrano, Daniel Gabriel Pons, Jorge Sastre-Serra

**Affiliations:** 1Grupo Multidisciplinar de Oncología Traslacional, Institut Universitari d’Investigació en Ciències de la Salut (IUNICS), Universitat de les Illes Balears, 07122 Palma de Mallorca, Spain; toni.martinez@uib.es (T.M.-B.); pere.morla@uib.es (P.M.M.-B.); lucas.melguizo@uib.es (L.M.-S.); mmunar@carrerasresearch.org (M.M.-G.); albamarotob@gmail.com (A.M.-B.); lida.torrens@uib.es (M.T.-M.); jordi.oliver@uib.es (J.O.); pilar.roca@uib.es (P.R.); mercedes.nadal@uib.es (M.N.-S.); jorge.sastre@uib.es (J.S.-S.); 2Instituto de Investigación Sanitaria de las Islas Baleares (IdISBa), Hospital Universitario Son Espases, Edificio S, 07120 Palma de Mallorca, Spain; 3Ciber Fisiopatología Obesidad y Nutrición (CB06/03), Instituto Salud Carlos III, 28029 Madrid, Spain

**Keywords:** tumorspheres, inflammation, chemoresistance, breast cancer and colorectal cancer

## Abstract

**Simple Summary:**

Chemoresistance poses a significant challenge in treating breast and colorectal cancers, making the identification of predictive biomarkers for chemotherapy response a crucial area of research. However, validating in vitro results can be difficult due to varied outcomes. This study investigates the use of 3D tumorspheres as an in vitro model to validate chemoresistance biomarkers in breast and colorectal cancer. The research emphasizes the role of inflammation-related pathways in influencing chemotherapy response. In silico analysis identified specific biomarkers that were elevated in patients who responded to treatment, and these markers were more prominent in 3D tumorspheres compared to traditional 2D cell cultures. Additionally, the study found that breast and colorectal cancer cells formed more tumorspheres in response to chemotherapy drugs (cisplatin and oxaliplatin, respectively), while cell viability decreased in standard adherent cultures. This suggests that the 3D tumorsphere model more accurately replicates the tumor environment and chemoresistance mechanisms. The findings support the use of 3D tumorspheres as a superior model for validating in silico data and studying the connection between inflammation and chemoresistance in breast and colorectal cancers.

**Abstract:**

Chemoresistance remains a major challenge in the treatment of breast and colorectal cancer. For this reason, finding reliable predictive biomarkers of response to chemotherapy has become a significant research focus in recent years. However, validating in vitro results may be problematic due to the outcome heterogeneity. In this study, we evaluate the use of tumorspheres as an in vitro model for validating biomarkers of chemoresistance in breast and colorectal cancer. Our investigation highlights the crucial role of inflammation-related pathways in modulating the response to chemotherapy. Using in silico approaches, we identified specific markers elevated in responders versus non-responders patients. These markers were consistently higher in three-dimensional (3D) tumorsphere models compared to traditional adherent cell culture models. Furthermore, the number of tumorspheres from breast and colorectal cancer cells increased in response to cisplatin and oxaliplatin treatment, respectively, whereas cell viability decreased in adherent cell culture. This differential response underscores the importance of the 3D tumorsphere model in mimicking the tumor microenvironment more accurately than adherent cell culture. The enhanced chemoresistance observed in the 3D tumorspheres model and their correlation of data with the in silico study suggest that 3D culture models are a better option to approach the in vivo model and also to validate in silico data. Our findings indicate that tumorspheres are an ideal model for validating chemoresistance biomarkers and exploring the interplay between inflammation and chemoresistance in breast and colon cancer.

## 1. Introduction

Cancer, characterized by intratumoral and intertumoral heterogeneity, remains a global health challenge with deep implications for patients and health systems [1]. The complex landscape of cancer is underscored by the diverse cellular and molecular variations within tumors and the distinctive features between cancer types. Breast and colorectal cancer particularly stand out due to their higher incidence rates and the challenges they present in terms of diagnosis, treatment, and overall patient prognosis [2].

Understanding the mechanisms underlying cancer progression and metastasis has become crucial in developing more effective therapeutic strategies. Recent research has highlighted the pivotal role of inflammation in the progression of cancer cells and their metastatic spread [3]. Inflammation, once considered a consequence of tumorigenesis, is now recognized as an active participant in promoting cancer progression [3,4]. The inflammatory mechanisms are mediated by interleukins and their receptors, with a special importance on interleukin-6 (IL6) and CXCL8 in the tumoral progression and angiogenesis activation [5,6]. On the other hand, other molecules, such as PPARγ, have anti-inflammatory properties inhibiting inflammation-related genes and NF-κB in different ways [7].

A growing issue in cancer treatment is the need to improve the efficacy of chemotherapy, particularly with platinum-derived drugs [8,9]. The development of resistance to these chemotherapeutic agents is a major obstacle to successful outcomes in cancer patients. Overcoming this resistance is imperative to improve treatment outcomes and patient survival rates [9]. To address this challenge, researchers are increasingly focusing on the intricate relationship between inflammation and chemotherapy resistance [10,11]. A better understanding of the interplay between inflammation and chemotherapy resistance could lead to the development of targeted therapeutic interventions, ultimately improving patient outcomes [12]. This knowledge could help identify key signaling processes and molecular players contributing to chemoresistance.

A valuable model for investigating chemoresistance in cancer is the use of tumorspheres. Tumorspheres, three-dimensional cultures derived from cancer cells, provide a more tumor-like representation of the in vivo tumor architecture compared to the traditional two-dimensional adherent cell culture [13,14,15]. These models allow researchers to study the dynamic interactions between cancer cells and inflammatory mediators in a more physiological context [13]. Tumorspheres are an invaluable tool for dissecting the complex mechanisms of chemoresistance, providing crucial insights for developing targeted and personalized therapeutic approaches [16].

In this research article, we explore the current understanding of inflammation and chemoresistance in breast and colorectal cancer. By using the tumorsphere model, we aim to increase the knowledge about these interconnected aspects that will ultimately guide the development of more effective and targeted therapeutic strategies for these prevalent cancer types.

## 2. Materials and Methods

### 2.1. Data Collection

The datasets GSE25066 and GSE28702 used for analysis in this study were obtained by searching the Gene Expression Omnibus (GEO; https://www.ncbi.nlm.nih.gov/geo; accessed on 3 December 2023) database and filtered by different criteria, including characteristics of data, experiment, and sample. Samples of identified datasets were filtered and classified to compare the Differentially Expressed Genes (DEGs) between platinum-based drug responders and non-responders in breast and colorectal cancer patients. Therefore, from all samples, data of 92 non-responder and 68 responder luminal A breast cancer patients, as well as 18 non-responder and 15 responder primary colorectal cancer patients, were used for Gene Set Enrichment Analysis (GSEA). Tissue samples were collected prior to any chemotherapy. Information on the inclusion and exclusion criteria of original cohorts is available in the GSE25066 and GSE28702 databases.

### 2.2. Gene Set Enrichment Analysis of Responders Versus Non-Responders Breast Cancer and Colorectal Cancer Patients

DEGs with a *p*-value ≤ 0.05 were used to obtain Normalized Enrichment Scores (NES) of Kyoto Encyclopedia of Genes and Genomes (KEGG) pathways using the GSEA method of WEB-based GEne SeT AnaLysis Toolkit (WebGestalt). Only KEGG pathways with a *p*-value ≤ 0.05 and a False Discovery Rate (FDR) value ≤ 0.05 were considered significantly enriched. The FDR is the proportion of false positives among all results identified as significant in multiple-hypothesis testing.

### 2.3. Reagents

Three-dimensional (3D) Tumorsphere Medium XF (3DTM) was obtained from Promocell (Heidelberg, Germany). Cisplatin (CDDP), Oxaliplatin (OXA), and TriReagent were purchased from Sigma-Aldrich (St. Louis, MO, USA). Primers were purchased from TIB MOLBIOL (Berlin, Germany) and Integrated DNA Technologies (Coralville, IA, USA) (Appendix A). Routine chemicals were supplied by Bio-Rad Laboratories (Hercules, CA, USA), Sigma-Aldrich (St. Louis, MO, USA), and Panreac (Barcelona, Spain). The 6-well and 96-well ultra-low attachment (ULA) plates were purchased from SPL Life Sciences (Pocheon-si, Republic of Korea).

### 2.4. Cell Culture

Human breast cancer cell line MCF7 and human colorectal cancer cell line SW620 were obtained from the American Type Culture Collection (ATCC; Manassas, VA, USA) and grown in 100 mm culture dishes in DMEM supplemented with 10% FBS (*v*/*v*) and 1% penicillin and streptomycin (*v*/*v*). Cells were maintained at 70–80% confluency and subcultured for further studies.

### 2.5. Tumorsphere Generation

For tumorsphere generation, MCF7 and SW620 single-cell suspensions were seeded at a density of 3 × 10^4^ cells/mL and 2 × 10^4^ cells/mL, respectively, in an ultra-low attachment 6-well plate (Promocell, Heidelberg, Germany) and cultured in 3DTM medium to allow anchorage-independent growth at 37 °C in a 5% CO_2_ humidified incubator. The formation of primary tumorspheres was determined after 48 h under an inverted microscope at 100× magnification.

### 2.6. RNA Isolation and RT-qPCR

Total RNA from MCF7 and SW620 tumorspheres were extracted by using Tri Reagent^®^ (Catalog no. T9424, Sigma-Aldrich) according to the manufacturer’s instructions. RNA concentration and purity were measured using a BioSpec-nano spectrophotometer (Shimadzu Biotech, Kyoto, Japan) set at 260 nm and 280 nm, obtaining a 260/280 nm ratio. cDNA was obtained by retrotranscription, and PCR reactions were carried out as previously reported [17]. Briefly, 1 μg of the total RNA was denatured at 90 °C for 1 min. Then, RNA was reverse transcribed to cDNA with 200 U MuLV reverse transcriptase in a 10 µL volume of retrotranscription reaction mixture containing 10 mM Tris-HCl (pH 9.0), 50 mM KCl, 0.1% Triton X–100, 2.5 mM MgCl_2_, 10 mM DTT, 2.5 µM random hexamers, 20 U RNase inhibitor, and 500 µM each dNTP. The mix was set at 25 °C for 10 min, 37 °C for 50 min, 70 °C for 15 min, and 4 °C until further processed. Each cDNA was diluted 1/10, and aliquots were frozen (−20 °C) until the PCR reactions were carried out.

PCR was performed using SYBR Green technology on a LightCycler 480 System II rapid thermal cycler (Roche Diagnostics, Basel, Switzerland). The genes, primers, and temperatures for the annealing step are specified in Table 1. The total reaction volume was 10 μL, containing 7.5 μL of SYBR Green Mix (containing SYBR green TB Green Premix ExTaq (Takara Bio, Shiga, Japan) and the sense and antisense specific primers for a final concentration of 0.2 µM and 2.5 µL of the cDNA template. The amplification program consisted of a preincubation step for denaturation of the template cDNA (5 min, 95 °C), followed by 45 cycles consisting of a denaturation step (10 s, 95 °C), an annealing step (10 s, temperature depending on primers; listed in Appendix A), and an elongation step (12 s, 72 °C min). A negative control lacking a cDNA template was run in each assay.

GenEx Standard Software 7.4.2 (Multi-DAnalises, Gothenburg, Sweden) was used to analyze the Cp values of the RT-PCR, normalizing with 18S, beta-2-microglobulin (B2M), beta-actin (ACTB), Hydroxymethylbilane Synthase (HMBS), Peptidylprolyl Isomerase A (PPIA), tata binding protein (TBP), and tyrosine 3-monooxygenase/tryptophan 5-monooxygenase activation protein zeta (YWHAZ) as housekeeping genes.

### 2.7. Cell Viability Assay

MCF7 and SW620 cells were seeded at densities of 10,000 cells/well and 20,000 cells/well, respectively, in a 96-well adherent plate. The following day, MCF7 cells were treated with 2.5, 5, 10, and 20 μM CDDP and SW620 cells with 1.25, 2.5, 5, 10, and 15 µM OXA for 48 h. After CDDP and OXA treatment, the number of viable cells was determined with the DNA binding dye Hoechst 33342 (Sigma-Aldrich, St. Louis, MO, USA), as previously described [18]. Fluorescence was measured using an FL×800 microplate fluorescence reader (BIO-TEK, Winooski, VT, USA) set at 350 nm of excitation wavelength and 455 nm of emission wavelength.

### 2.8. Tumorsphere Formation Efficiency and Size Determination

To evaluate tumorspheres forming efficiency (TFE), single cells from adherent culture were seeded at a density of 10,000 cells/well of MCF7 and 5000 cells/well of SW620 in ultra-low attachment 96-well plates with increasing concentrations of CDDP (2.5, 5, 10 and 20 μM) and OXA (1, 2.5, 5 and 10 μM), respectively, for 48 h. Tumorspheres forming efficiency (TFE) was calculated based on the following ratio: TFE (%) = (number of tumorspheres formed per well)/(number of cells seeded per well) × 100. After the culture period, tumorspheres larger than 80 μm were counted under an inverted microscope at 100× magnification. Well fields were digitally imaged and tumorsphere size determination was performed by area measurements using the ImageJ2 software (Bethesda, ND, USA).

### 2.9. ROC Analysis

Identified biomarkers were analyzed using the ROCplotter (www.rocplot.org, accessed on March 2024) on the transcriptome data of a large set of breast cancer patients (*n* = 475) and colorectal cancer patients (*n* = 805) to compare the expression levels of biomarkers between responders and non-responders patients. ROC curve with *p*-value ≤ 0.05 was considered significant between the two groups to evaluate the prediction ability of the identified genes [19].

### 2.10. Statistical Analysis

Statistical Program for the Social Sciences software for Windows (SPSS, version 24.0; SPSS Inc., Chicago, IL, USA) was used for all statistical analysis. Data are expressed as mean values ± standard error of the mean (SEM). Differences among experimental groups were statistically analyzed using Student’s *t*-test. Statistical significance was defined as *p*-values *p* ≤ 0.05.

## 3. Results

### 3.1. Identification of Main Chemotherapy-Resistance Pathways Involved in Breast and Colorectal Cancer

To identify pathways related to chemotherapy resistance in breast and colorectal cancer, the GSEA software v0.4.X (Vancouver, BC, Canada) was used to analyze KEGG pathway enrichment of all DEGs (|Fold Change (FC)| ≥ 1 and a *p*-value ≤ 0.05) between responders and non-responders to chemotherapy. As shown in Figure 1A, breast cancer patients who responded efficiently to chemotherapy showed a negative NES in KEGG pathways that included *Primary immunodeficiency*, *Phagosome*, *Chemokine signaling pathway*, *Cytokine-cytokine receptor interaction*, and *Toll-like receptor signaling pathway*. On the other hand, colorectal cancer patients who responded efficiently to chemotherapy presented a negative NES in KEGG pathways such as *IL-17 signaling pathway*, *TNF signaling pathway*, *Chemokine signaling pathway*, *Cellular senescence*, and *Cytokine-cytokine receptor interaction* (Figure 1B). For those common pathways between both breast and colorectal cancer, an enrichment plot was performed, showing that *Cytokine-cytokine receptor interaction* and *Chemokine signaling* pathways were highly statistically significant (Breast cancer-*Cytokine-cytokine receptor interaction*: *p*-value ≤ 0.001, FDR ≤ 0.001; Breast cancer-*Chemokine signaling*: *p*-value ≤ 0.001, FDR ≤ 0.001; Colorectal cancer-*Cytokine-cytokine receptor interaction*: *p*-value ≤ 0.001, FDR ≤ 0.001; Colorectal cancer-*Chemokine signaling*: *p*-value ≤ 0.001, FDR ≤ 0.001), as observed in Figure 1A,B.

In addition, to further explore both *Cytokine-cytokine receptor interaction* and *Chemokine signaling* pathways, the size and number of leading-edge IDs and genes are described in Table 1, highlighting the relevance of some of the genes involved in these pathways, such as CXCL8, NFKB1, and IL6.

### 3.2. Tumorsphere Generation Enhances Cytokine and Chemokine Pathways

To assess whether tumorsphere 3D culture accurately mimics the physiological scenario, the mRNA expression of genes related to *Cytokine-cytokine receptor interaction* and *Chemokine signaling* pathways was analyzed in breast cancer (MCF7) and colorectal cancer (SW620) tumorspheres in comparison to adherent cell culture. As shown in Figure 2A, breast cancer tumorspheres presented a statistically significant increase in CXCL8, IL6, IL6R, and NFKB1 mRNA levels in comparison to adherent cell culture, while TGFB1 mRNA expression was decreased. Similarly, colorectal cancer tumorspheres showed increased mRNA levels of CXCL8 and IL6R. Although data on IL6 mRNA levels are not shown due to undetectable expression in the SW620 cell line, PPARG and TGFB1 mRNA levels decreased after colorectal cancer tumorsphere generation (Figure 2B).

To evaluate the mRNA expression levels of genes related to *Cytokine-cytokine receptor interaction* or *Chemokine signaling* pathways in responders and non-responders breast and colorectal cancer patients, the ROCplotter bioinformatic tool was used. As presented in Figure 3A, CXCL8 (*p* = 0.12), IL6 (*p* = 0.022), and NFKB1 (*p* = 0.0097) mRNA levels were higher in breast cancer patients who did not respond to chemotherapy treatment. Additionally, TFGB1 (*p* ≤ 0.001) mRNA levels were lower in these patients. The signature profile analysis of CXCL8, IL6, and NFKB1 revealed a higher mRNA expression in non-responder breast cancer patients (*p* = 0.015). On the other hand, in colorectal cancer patients who did not respond to chemotherapy treatment, CXCL8 (*p* = 0.018), IL6 (*p* = 0.01), and IL6R (*p* ≤ 0.001) mRNA levels were higher, while TFGB1 (*p* ≤ 0.001) mRNA levels were lower, as observed in Figure 3B. The signature profile analysis of CXCL8, IL6, and IL6R revealed a significantly elevated mRNA expression in non-responder colorectal cancer patients (*p =* 0.099).

### 3.3. Chemotherapy Treatment Shows Less Effectiveness in Breast and Colorectal Cancer Tumorspheres

To investigate the effect of chemotherapy treatment on breast and colorectal cancer tumorspheres compared to adherent cell culture, cell viability in adherent cell culture, tumorsphere formation efficiency (TFE), and sphere area analysis were assessed. As shown in Figure 4A,E, MCF7, and SW620 cells showed a strong decrease in cell viability with increasing doses of platinum-derived drugs (CDDP and OXA, respectively). Nevertheless, despite observing a decrease in tumorsphere area in breast and colorectal cancer tumorspheres (Figure 4C,D,G,H), both showed an increase in TFE after chemotherapy treatment (Figure 4B,F).

## 4. Discussion

This study highlights the potential of tumorspheres as a promising model for identifying markers of treatment resistance. Tumorspheres, characterized by their three-dimensional structure and ability to mimic some aspects of tumor biology, provide a valuable tool for studying cancer biology and therapeutic responses. A key feature of this model is the release of cytokines and chemokines, which play a pivotal role in creating an inflammatory microenvironment that supports tumor growth and resistance mechanisms [3,4]. The search for markers of treatment resistance is crucial in the continuous effort to manage cancer. While validating these markers through in vitro analyses is particularly relevant, this approach presents significant challenges, such as the complexity of generating organoids and the limitations of traditional adherent cultures that do not mimic in vivo conditions [13,14,15].

Despite significant advances in the treatment of breast and colorectal cancer, recurrence and drug resistance are still prevalent issues [20,21]. Inflammation, a well-established driver of tumor progression and a hallmark of cancer is emerging as a critical factor in cancer chemoresistance, acting through multiple mechanisms that promote tumor survival and reduce treatment efficacy [10,11,22]. Consistent with existing literature, our study underscores the dysregulation of inflammation signaling pathways in breast and colorectal cancer patients who fail to respond to treatment [3,23]. Specifically, the “*cytokine-cytokine receptor interaction*” and “*chemokine signaling pathway*” gene sets are prominently upregulated in both non-responders breast and colorectal cancer patients. We have focused on these two gene sets for their potential relevance in identifying common mechanisms of chemoresistance in both cancer types. Analyses of altered genes of these specific pathways highlight the significance of IL-6 and IL-8, which are pivotal in cancer biology [24,25]. IL-6 contributes to cancer chemoresistance by activating signaling cascades that promote cell survival and proliferation [24,26]. In addition, IL-6 modifies the tumor microenvironment and facilitates epigenetic changes, further enhancing resistance mechanisms [24,27]. Similarly, IL-8 interferes with the chemotherapeutic responses through analogous processes, promoting cancer cell survival and growth, recruiting immune cells that may protect tumor cells from the effects of chemotherapy, upregulating drug efflux pumps, and altering the tumor microenvironment [25,28]. Furthermore, our data indicated that breast cancer patients who do not respond to chemotherapy present higher IL-6 mRNA expression levels, while non-responder colorectal cancer patients have higher IL-6 and IL-8 mRNA expression levels. The upregulation of the cytokine-cytokine pathway has been identified as a contributing factor to chemoresistance rather than a result of it [23,26]. Remarkably, these cytokines, which were upregulated in non-responder breast and colorectal cancer patients, were also upregulated in breast and colorectal tumorspheres. This finding evidences the relevance of tumorsphere models in reflecting tumor biology and highlights their potential as robust tools for studying mechanisms of chemoresistance. Also, those cytokines that were upregulated in the tumorspheres in each cancer type were analyzed as a signature profile expression between responder and non-responder patients in each cancer type. Importantly, the signature profile applied to breast and colorectal cancer further reinforces this observation by demonstrating that the upregulated mRNA expression patterns in tumorspheres closely align with those observed in non-responder patients.

Tumorsphere culture has emerged as a promising approach in cancer biology research, though its application for biomarker validation is still unclear [13]. Also, there is a lack of knowledge of whether the cytokine profile expression could predict the tumor response to chemotherapy. Our findings obtained from tumorspheres not only correlate with chemotherapy resistance-related datasets but also demonstrate in vitro evidence of chemoresistance in cisplatin-treated breast-derived tumorspheres and oxaliplatin-treated colorectal-derived tumorspheres. Platinum-based drugs are widely known for their use in advanced and more resistant stages of cancer [29,30]. Nonetheless, cancer cells overcome chemotherapy treatment through various mechanisms, such as triggering inflammatory pathways [3,23]. As shown in our results, the adherent cell culture model exhibited higher sensitivity to cisplatin and oxaliplatin treatment, while the tumorspheres increased their sphere-formation efficiency. These data confirm that 3D cell culture has been successfully used as a chemotherapy resistance model. Our results suggest that the tumorspheres model may be a promising option for validating resistance-related biomarkers in breast and colorectal cancer cell lines.

## 5. Conclusions

This study highlights the potential of tumorspheres as a valuable model for identifying markers of treatment resistance in cancer. By closely mimicking tumor biology and the inflammatory microenvironment, tumorspheres provide a robust tool for studying cancer progression and therapeutic response. The tumorsphere model shows a strong correlation with chemotherapy resistance datasets, validating its effectiveness in identifying biomarkers associated with chemoresistance. Despite some challenges, the tumorsphere model’s ability to simulate in vivo conditions makes it a promising and powerful tool for analyzing predictive treatment-response markers.

## Figures and Tables

**Figure 1 biology-13-00724-f001:**
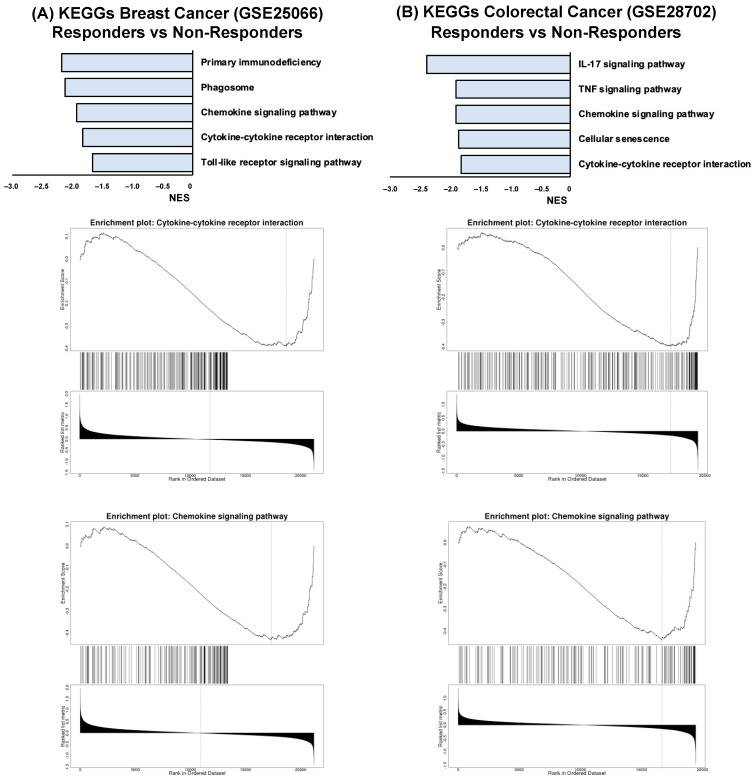
Identification of signaling pathways alterations in responders vs. non-responders breast (**A**) and colorectal (**B**) cancer patients. Pathways with FDR value ≤ 0.05 are represented. Enrichment plots of matched KEGG pathways are represented, respectively.

**Figure 2 biology-13-00724-f002:**
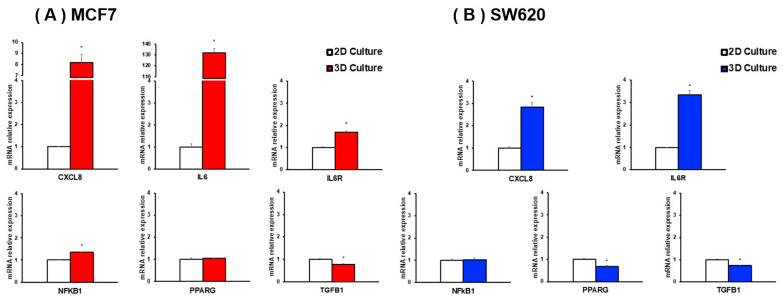
mRNA expression levels of inflammation-related markers in MCF7 (**A**) and SW620 (**B**) tumorspheres compared to a two-dimensional (2D) culture. Statistical significance was analyzed by Student’s *t*-test and set at * *p* ≤ 0.05.

**Figure 3 biology-13-00724-f003:**
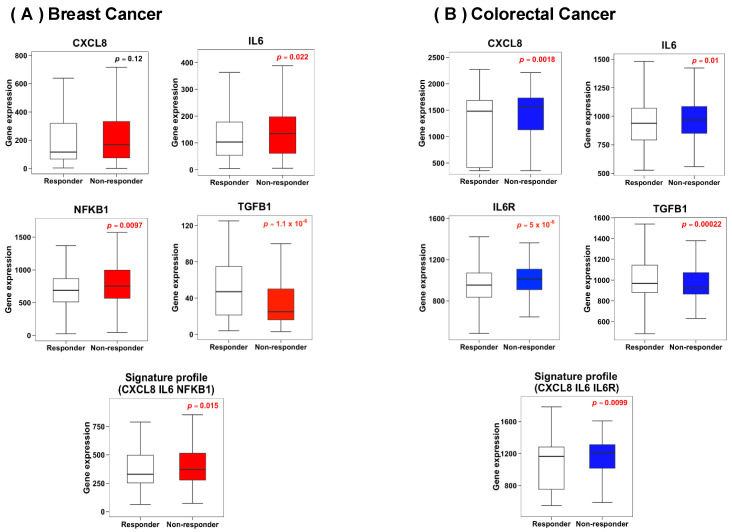
Gene expression of inflammation markers in breast cancer (**A**) and colorectal cancer (**B**) patients according to response to chemotherapy treatment. Statistical significance was analyzed by Student’s *t*-test and set at *p* ≤ 0.05 (highlighted values).

**Figure 4 biology-13-00724-f004:**
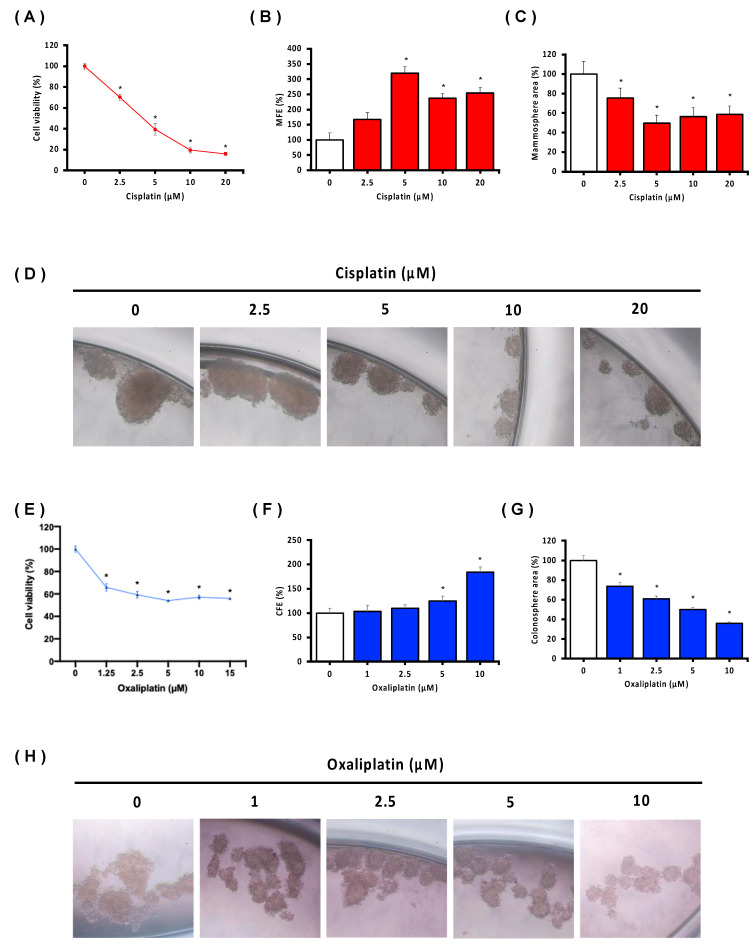
Viability and tumorsphere formation efficiency of MCF7 (**A**–**D**) and SW620 (**E**–**H**) cell lines after Cisplatin/Oxaliplatin treatment, respectively. Statistical significance compared to 0 µM (Cisplatin or Oxaliplatin) was analyzed by Student’s *t*-test and set at * *p* ≤ 0.05.

**Table 1 biology-13-00724-t001:** Detailed enriched KEGG pathways of responders vs. non-responders comparison in both breast and colorectal cancer.

KEGG	Breast Cancer (GSE25066)			Colorectal Cancer (GSE72970)		
	Size	Edges	Genes	Size	Edges	Genes
Chemokine signaling pathway	171	67	RHOA, RAP1B, GNAI2, GRK2, GNG10, GSK3A, RAP1A, LYN, PRKACB, GRK6, GRK6, CXCL8, ARRB2, ADCY7, PIK3CD, CCL4, CXCL10, CCL5, NCF1, CCR1, CCL3L1, CCL3L3, CCL3, CX3CR1, CXCR5, VAV1, SHC3, CCR7, XCL1, PIK3CG, PF4, ADCY8, GNG7, CXCR6, CCR2, CCR5, CXCR2, CXCR1, GNG5, PRKACG, CCR9, CCR2, CCL22, CCL17, HCK, CCR8, CCL7, CCR3, IKBKB, PRKCB, CCL11, CCL15, LYN, GRK6, IKBKB, CXCL11, PIK3CD, ITK, CXCR6, CXCL8, GRK6, CXCR4, DOCK2, RAC2, CCL8, NCF1, PPBP, RASGRP2, XCL2, XCL1, CXCL5, CCL2, CXCR5, CXCR4, ARRB1, CCL24, IKBKG, GRK2, WAS, GSK3A, STAT1	184	52	CXCL8, GNAI1, CCL26, CXCR2, CXCL1, CCL20, CCL8, CCL28, PF4, CCL3L3, PPBP, CXCL6, CXCL5, GRK3, FGR, CCR10, CXCL3, CCL4, ELMO1, HCK, GNAI2, NRAS, NFKB1, RAC2, PAK1, CCR1, CXCL2, NCF1, PRKCD, MAPK1, AKT1, CCR3, GNB1, MAP2K1, PIK3CG, GNG10, CXCL9, RAF1, GNAI3, CX3CL1, CCR2, PRKACB, LYN, CXCR4, GRK5, KRAS, GNG2, CCL13, GNG11, STAT3, CCL25, GRB2
Cytokine-cytokine receptor interaction	257	69	PPBP, IL1R2, CXCR2, CXCR1, TNFRSF10C, CCR1, PF4, TNFRSF10C, IL18, IL4, CCR3, IL2RA, IL24, LTB, ACKR4, IL16, IL32, CCR1, TNFRSF25, CCR7, CCL3L1, CCL3L3, CCL3, CCL8, CX3CR1, CD4, TNFSF8, IL13RA2, XCL2, XCL1, CXCL11, IL36RN, CD27, IL1R2, IL21R, IL7R, XCL2, XCL1, TNFRSF25, CSF2RA, IL21R, TNFRSF25, CCL2, TNFRSF10C,IL1B, CRLF2, IL11, CSF2RB, CCL4, IL2RG, CXCR4, CCL17, CD70, TNFSF12, CSF3R, TNFRSF17, TNFRSF11A, CXCR5, IL1RAP, IL16, IL11RA, CCL5, TNFSF18, CCL15, IL9R, CCL7, IL10RA, TNFRSF14, LTA, CCR5, TNFRSF1B, IL21, TNFSF14, CXCR5, BMP3, IL10, CSF3, IL37, CCL5, IL18RAP, CXCL8, IL11, TNFRSF25, IL1B, IL4, CXCR4, IL9R, CXCR4, XCL1, IL1RAP, CSF2RA, IL2RA	284	58	CXCL8, IL1B, CCL26, CXCR2, CSF2RB, CXCL1, IL24, CCL20, CCL8, CCL28, IL13RA2, PF4, CCL3L3, TNFSF9, TNFRSF11A, FAS, PPBP, IL13RA1, TNFRSF10D, IL6, IL17RB, CXCL6, TNFRSF17, CXCL5, IL18R1, INHBB, CCR10, CXCL3, CCL4, TNFRSF10A, TNFRSF21, CSF3R, IL15, TNFRSF18, CD27, BMP4, OSM, TNFRSF11B, IFNGR2, BMP2, CCR1, CXCL2, GHR, IL1R2, CCR3, CXCL17, ACKR4, TNFSF18, IL7R, IL18, TNFRSF10B, PRL, IL2RA, CXCL9, CX3CL1, CCR2, TGFBR2, IL10

## Data Availability

The data presented in this study are available on request from the corresponding author.

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
