# Peer review of "Tumorspheres as In Vitro Model for Identifying Predictive Chemoresistance and Tumor Aggressiveness Biomarkers in Breast and Colorectal Cancer"

_biology, 2024, doi:10.3390/biology13090724_

Round 1
Reviewer 1 Report
Comments and Suggestions for Authors
In this manuscript, the authors reported study using tumorspheres as an in vitro model to identify predictive biomarkers for chemoresistance and tumor aggressiveness in breast and colorectal Cancer. I have following comments.
1. Please elaborate materials and methods section.
2. Include the inclusion and exclusion criteria for sample selection in the materials and methods section.
3. What was the rationale behind selecting the specific concentrations of CDDP and OXA? Are these concentrations comparable to the levels used in chemotherapy?
4. Please add blue bars to Figure 1B.
5. Why did you choose to focus on the mRNA expression of genes related to Cytokine-cytokine receptor interaction and Chemokine signaling pathways, when other signaling pathways have more negative NES according to Figure 1?
6. Clarify why tumor-forming efficiency increases while cell viability decreases after treatment.
7. Did you assess the cytokine profile of tumorspheres following treatment with oxaliplatin and cisplatin?
8. Did you observe any differences between male and female subjects?
Comments on the Quality of English LanguageDear Editor,
In conclusion, I believe this manuscript has the potential to make a significant contribution to the field with revisions. I recommend a major revision to address the concerns above and enhance the overall quality of the manuscript.
Thank you for the opportunity to review this manuscript. I look forward to seeing the authors' revisions and the final decision on its publication.
Best regards,
Rekha Jalandra
Reviewer 2 Report
Comments and Suggestions for Authors
The manuscript is not well-written in terms of explaining the significance of the study. Several aspects of the manuscript need to be improved for it to meet the standards of publication.
Specific Comments:
-
The introduction lacks sufficient background on the relationship between inflammation and cancer. I recommend that the authors include references to relevant studies or reports that explore this connection. This would provide a stronger foundation for the study and help contextualize the research within the broader scientific landscape.
-
The manuscript does not explain what FDR (False Discovery Rate) is or how it is applied in the context of the study. The authors should define FDR and describe its relevance and application in the analysis of their data. This is crucial for the readers to understand the statistical approach used in the study.
-
The authors have not mentioned the specific cDNA synthesis and qPCR kits used in the study. This information is essential for reproducibility and should be included in the Methods section.
-
The authors have stated that a >1-fold change is considered significant, but in most studies, a >2-fold change is typically used to denote significant upregulation. The authors should provide a reference to support their choice of a >1-fold change threshold and explain why this criterion was selected for their analysis.
-
The Results section is poorly written and lacks depth. It appears that the authors have merely listed their findings without discussing the significance of the results in each section. The Results should be rewritten to include a detailed explanation of the significance of the findings, discussing how they contribute to the overall objectives of the study and the implications they may have for the field.
-
In the Discussion section, the authors describe their approach as novel. However, this is not accurate, as there are many published reports using similar study approaches. I recommend that the authors remove the word "novel" from the manuscript, as it does not accurately reflect the study's contribution. Instead, they should focus on how their findings add to the existing body of knowledge.
While the manuscript addresses an important topic, it requires significant revision to improve clarity, provide necessary details, and accurately reflect the contribution of the study. I recommend that the authors address the specific comments outlined above and revise the manuscript accordingly.
Comments on the Quality of English LanguageThe manuscript is well-written and the quality of the English language is generally good. There are a few minor grammatical and typographical errors present throughout the text, but these can be easily corrected with careful proofreading. Overall, the language is clear and effectively conveys the intended information.
